# Natural Carbon Nanodots: Toxicity Assessment and Theranostic Biological Application

**DOI:** 10.3390/pharmaceutics13111874

**Published:** 2021-11-05

**Authors:** Ming-Hsien Chan, Bo-Gu Chen, Loan Thi Ngo, Wen-Tse Huang, Chien-Hsiu Li, Ru-Shi Liu, Michael Hsiao

**Affiliations:** 1Genomics Research Center, Academia Sinica, Taipei 115, Taiwan; ahsien0718@gate.sinica.edu.tw (M.-H.C.); g803020178g@gate.sinica.edu.tw (C.-H.L.); 2Department of Chemistry, National Taiwan University, Taipei 106, Taiwan; r09223161@ntu.edu.tw (B.-G.C.); d10223121@ntu.edu.tw (L.T.N.); d09223104@ntu.edu.tw (W.-T.H.); 3Nano Science and Technology Program, Taiwan International Graduate Program, Academia Sinica and National Taiwan University, Taipei 115, Taiwan; 4Department of Biochemistry, College of Medicine, Kaohsiung Medical University, Kaohsiung 807, Taiwan

**Keywords:** natural carbon nanodots, tumor-targeting probes, biosensing, cancer theranostic, toxicity assessment

## Abstract

This review outlines the methods for preparing carbon dots (CDs) from various natural resources to select the process to produce CDs with the best biological application efficacy. The oxidative activity of CDs mainly involves photo-induced cell damage and the destruction of biofilm matrices through the production of reactive oxygen species (ROS), thereby causing cell auto-apoptosis. Recent research has found that CDs derived from organic carbon sources can treat cancer cells as effectively as conventional drugs without causing damage to normal cells. CDs obtained by heating a natural carbon source inherit properties similar to the carbon source from which they are derived. Importantly, these characteristics can be exploited to perform non-invasive targeted therapy on human cancers, avoiding the harm caused to the human body by conventional treatments. CDs are attractive for large-scale clinical applications. Water, herbs, plants, and probiotics are ideal carbon-containing sources that can be used to synthesize therapeutic and diagnostic CDs that have become the focus of attention due to their excellent light stability, fluorescence, good biocompatibility, and low toxicity. They can be applied as biosensors, bioimaging, diagnosis, and treatment applications. These advantages make CDs attractive for large-scale clinical application, providing new technologies and methods for disease occurrence, diagnosis, and treatment research.

## 1. Introduction

The materials studied in the past were at the micrometer scale; however, in the past few decades, nanometer-scale development has changed, which has become an essential direction of scientific and technological development. Nano-sized materials are widely used in various fields, such as medicine, biosensor development, energy research, and catalysis. As fluorescent nanomaterials can emit light, they have the potential for application in biomarking technology, such as semiconductor-based quantum dots containing cadmium sulfide (CdS) or cadmium selenide (CdSe). However, most materials with a high quantum yield (QY, ϕ) on the market include heavy metal components, which pose risks related to their biological toxicity and cytotoxicity. Contamination may occur during the synthesis process, and they are not adequately recycled after use; as such, the associated harm to the environment should not be underestimated. The above shortcomings limit the application range as semiconductor quantum dots.

In recent years, nanotechnology has been widely used in the field of biomedicine [1]. Generally, when the scale of a substance drops to the nanometer level, its physical and chemical properties also undergo tremendous changes, especially in terms of its optical properties, providing an ideal development space for applications in the field of biomarkers and optical imaging of diseases [2]. Since their inception, carbon nanomaterials have attracted attention from researchers in materials science and biomedicine, especially considering their excellent optical properties, facilitating their application in bioimaging, biomarkers, and sensors [3]. Fluorescent carbon dots (CDs) are another new type of carbon nanomaterial, with a typical particle size of about 10 nm. In addition to the advantages related to the particle size and wavelength-dependent luminescence characterizing quantum dots, fluorescent CDs also have high light stability and present no light scintillation phenomena. Their surface is easily functionalized and modified, and the preparation materials are widely available [4,5,6]. Wang et al. evaluated the toxicity of CDs and showed that they do not cause any abnormalities or damage to tissues and organs [7]. Kang et al. compared CDs, single-walled carbon tubes, and carbon dioxide [8]. After the cytotoxicities of silicon and zinc oxide were determined, CDs showed the lowest toxicity compared with the materials mentioned above. CDs’ unique optical properties and excellent biological properties confer outstanding advantages and good development prospects in biomedical optical imaging and tumor diagnosis. In addition, the ability to introduce multiple functional groups onto the surface of CDs also provides possibilities for their modification.

Green manufacturing is a modern manufacturing model that comprehensively considers the environmental impact and resource benefits. Due to the increasing awareness of environmental protection, green manufacturing is becoming an increasingly important process in all walks of life. The chemical synthesis of CDs can be similar to that produced from various natural resources in the nanoscale process. For instance, Hsu et al. used coffee grounds to synthesize CDs by calcining in 2012. The average particle size of the synthesized CDs was about 5 nm, and they were successfully applied to biological imaging. Unlike the previous need for complex processing procedures, a rapid and straightforward synthesis method was developed. Later, Crista et al. used different organic compounds to synthesize CDs, with an average particle size of about 2.6–7.9 nm, with many carboxyl and amine groups, proving that the quantum yield varies with functional groups [9]. In 2013, Qu et al. used citric acid and ethylenediamine to synthesize CDs, leading to a quantum yield as high as 60.2% [10]. It can be speculated that naturally synthesized carbon nanomaterials should have higher biocompatibility and lower toxicity. To date, many techniques for synthesizing CDs from natural sources have been developed, including laser ablation, arc discharge, electrochemistry, thermal decomposition, ultrasonic, and microwave methods, among others. In particular, the thermal decomposition method is simple, safe, fast, and effective. Research has found that carbon nanoparticles synthesized by this method have good fluorescent performance, although the quantum yield is not high [11,12,13]. The optical properties, stability, and biocompatibility are still powerful enough for use in bioapplications. This review introduces and simplifies the current carbon nanocomposite synthesis techniques. We provide a brief introduction to fluorescent CDs, mainly reviewing their structural characteristics, carbon source materials, preparation methods, luminescence mechanism, and applications in the field of biomedicine. Obtaining CDs from natural resources can produce nanomaterials that are more environmentally friendly. The considered technique intends to use available (in a daily sense) materials to synthesize low-toxicity fluorescent nanomaterials. The production process does not require the use of many precursor chemicals to build natural CDs. We hope that developing a safe and straightforward production process can unite fluorescent materials and natural CDs and contribute to understanding and exploring the application of nano-fluorescent materials.

## 2. Natural Carbon Nanodots

CDs are a new type of carbon nanomaterials with luminescence characteristics, quasi-zero dimension, relatively simple preparation, abundant sources of raw materials, easy surface functionalization, low toxicity, and good biocompatibility. The fluorescence wavelength can be adjusted, and the two-photon absorption area is large. In various literature reports, CDs have also been referred to as carbon quantum dots (CQDs), carbon nanodots (CNDs), Graphene quantum dots (GQDs), carbon nanocrystals, and so on. Due to their excellent performance, CDs show promising potential application prospects in sensing, biology, medicine, food, environment, catalysis, photoelectricity, energy, and so on. Researchers have conducted extensive scientific studies and made significant progress [14]. CDs synthesis methods can be mainly divided into two categories: One uses physical or chemical means to crack larger carbon structures (e.g., carbon nanotubes, graphite rods, graphene, carbon powder) into tiny CDs; these top-down methods include arc discharge methods, laser ablation methods, chemical oxidation methods, and so on. Small organic molecule precursors, such as sugar, citric acid, and amino acids, are used as carbon sources for these methods through functional group coupling to achieve polymerization to prepare the CDs [15,16,17]. The second category comprises bottom-up methods, such as electrochemical, hydrothermal, pyrolysis, microwave-assisted, and ultrasonic methods [18].

Other methods, such as template methods, neutralization reaction exothermic methods, and micro-fluidized bed technology methods, can also be used to prepare carbon dots [19,20,21]. The preparation of CDs well-embodies the concept of green chemistry, using cheap, environmentally friendly carbon source precursors and natural renewable, cheap raw materials as carbon sources. The resources for synthesizing CDs can be found in the natural environment, such as eggs, grass, tea leaves, leaves, silk, silkworm pupae, shrimp shells, grapefruit peel, peanut shells, coffee grounds, beer, and other materials. CDs were discovered and debuted for the first time due to their fluorescent luminescence characteristics. The luminescence mechanism of CDs has always been a critical research direction for researchers, considering factors such as the quantum size effect, surface state, functional group mechanisms, electron holes, and radiation. Their rearrangement theory has been studied in various aspects. Although a complete theoretical explanation system of the CDs fluorescence mechanism has not been formed, the absorption and fluorescence of CDs exhibit properties such as photoluminescence, chemiluminescence, electrochemiluminescence, and luminescence. The conversion of photoluminescence, peroxidase-like activity, non-toxicity, and other physical and chemical properties provides a solid and feasible theoretical basis for further research.

The main application research directions of CDs can be divided into the following categories (Figure 1):Imaging: multicolor fluorescent images of mammalian cells, plant cells, and micro-organisms, and imaging in mice;Photocatalysis: the degradation of organic molecules, the reduction in CO_2_, and water splitting;Optoelectronic devices: LEDs and solar cells (sensitizer/co-sensitizer, transport layer, electrolyte, and/or co-catalyst for counter electrode);Sensors: food quality and safety, drug analysis, environmental pollution determination, immunoassay, and other fields, such as detecting heavy metal ions, anions, pesticides, molecules, small organic molecules, and/or nucleic acids;Electrocatalysis: mainly used in oxidation-reduction reactions, oxygen evolution reactions, hydrogen evolution reactions, and reduction reactions for carbon dioxide, and dual-function catalysts;Biomedicine: photodynamic therapy, photosensitizers for cancer cell destruction, radiotherapy, the tracing and delivery of drugs or genes, drug release, and anticancer drugs.

Most carbon sources derive from exhausted petroleum resources or non-environmentally friendly manufacturing processes. Therefore, using recycled materials and bio-renewable resources to develop high-performance CDs is a critical green environmental issue. In recent years, the awareness of environmental protection has significantly risen, and, as such, naturally derived CDs have gradually received more attention.

The main factor is that they are derived from renewable and sustainable biological materials; for example, lignocellulose from dead wood, waste wood, rice straw, bagasse, wheat straw, etc. Suppose that such a raw material can be used as a carbon resource and converted into CDs with therapeutic and diagnostic value. In that case, we can imagine the result will not overwhelm the demand of the food supply chain. Still, it can also address waste disposal problems while applying the resource to produce high value-added products such as electronics, energy, and biomedicine. Table 1 summarizes the existing research on converting various natural resources into CDs, and the associated applications and emission spectra are described in detail.

## 3. Toxicity Evaluation of Natural Carbon Nanodots

CDs can be derived from a wide range of synthetic raw materials, low-cost, stable chemical properties, and non-toxic materials. The application of CDs in medicine and pharmacy was recently extensively studied. One of the most promising nanomaterials is carbon quantum dots (CQDs). For this purpose, the toxicity of CQDs was investigated in cells and living systems (Table 2). In 2019, L. Janus used human dermal fibroblasts to conduct a cytotoxicity study of N-doped chitosan-based CDs [129]. As shown in Figure 2, after 48 h, the cell viability was recorded as 94%. CDs were synthesized by utilizing non-toxic raw materials and removing unreactive residues during purification.

First, we start with the selection of carbon source and the adjustment of dopants that synthesize carbon dots, and at the same time list the test models of carbon dots. Secondly, according to the selective response of the carbon dot to the biological models, the corresponding toxicity was constructed, and the actual application was investigated. As mentioned in Table 2, CQDs are usually doped with nitrogen to enhance their fluorescence quantum yield and optical performance [130,131]. CDs doped with nitrogen (such as carbon nanotubes, graphene, hollow spheres, etc.) have unique properties. They can inject electrons into the carbon substrate to change the electron and transport characteristics such as sensors, nanogenerators, etc., and are widely used and have become a research trend. Moreover, green methods that utilize natural biomass/biowaste and micro-organisms without introducing toxic chemicals as CDs precursors have been widely used to synthesize CQDs [132].

**Table 2 pharmaceutics-13-01874-t002:** Toxicity evaluation methods for CDs.

Material	Sources	Concentration	QY (%)	Cells or Animal Models	Toxicity	Ref.
Carbon quantum dots	Medicinal mulberry leaves	500 ug/mL	9.7	Human normal hepatic stellate cell line LX-2 cells and human HCC cell line HepG2 cells	Almost non-cytotoxic	[133]
Carbon dots	Mango peel	500 ug/L	8.5	A549 cells	Remained above 90% Low toxicity	[134]
Nitrogen-doped carbon quantum dots	Watermelon juice	300 ug/mL	10.6	HepG2 cells	Remained 90% Low cytotoxicity	[135]
N-doped carbon quantum dots	Bio-waste lignin	100 mg/mL	8.1	Mouse macrophage cells	Remained 96.8% Low toxicity	[136]
Carbon dots	Roast duck	1 mg/mL	38.05	PC12 cells and *C. elegans*	Remained 91.19% Low toxicity	[122]
Nitrogen-doped carbon dots	*P. acidus* fruit juice	200 ug/mL	12.5	Cells and *C. elegans*	Remained 93% Low cytotoxicity	[137]
Carbon quantum dots	*Salvia* *hispanica L*. seeds	250 ug/mL	17.8	HEK293 cell line	Remained 91.7% Low toxicity	[138]
Carbon dots	Wheat straw	0.8 mg/mL	7.5	HeLa cells	Negligible cytotoxicity	[139]
Carbon dots	*Malus floribunda* fruit	200 ug/mL	19	Cells and *C. elegans*	Remained 93% and low toxicity	[140]
Carbon quantum dots	Banana peel waste	200 ug/mL	20	*C. elegans*	Low toxicity	[141]
Nitrogen and sulfur dual-doped carbon quantum dots	Fungus fibers	400 ug/mL	28.11	HepG2 cells	Remained over 95% Low cytotoxicity	[142]
Carbon dots	Sweet lemon peel	500 ug/mL	n/a	MDA-MB231 cells	Remained above 75% Low cytotoxicity	[143]
Carbon dots	Lychee waste	1.2 mg/mL	23.5	Skin melanoma cells	Remained above 89% Low cytotoxicity	[144]
Nitrogen-doped carbon quantum dots	Citrus lemon	2 mg/mL	31	Human breast adenocarcinoma cells	Remained above 88% low cytotoxicity,	[145]
Carbon dots	*Daucus carota subsp. sativus* roots	1 mg/mL	7.6	MCF-7 cells	Remained above 95% Low toxicity	[146]
Carbon nanodots	Custard apple peel waste biomass	100 ug/mL	n/a	HeLa and L929 cells	Remained above 85% Low toxicity	[147]
Carbon quantum dots	Pineapple peel	1 mg/mL	42	HeLa and MCF-7 cells	Remained 84% Low toxicity	[148]
N-carbon dots	Jackfruit seeds	2 mg/mL	17.91	A549 cells	Remained 96% and less toxic	[149]

S. Cong et al. [122] used PC12 cells for a cytotoxicity study of CQDs obtained from a roast duck. After 36 h, the cell viability of PC12 was recorded as 91.19% at the concentration of 1 mg/mL. In addition, using CDs at a concentration of 15 mg/mL for treating nematodes did not lead to any death for 24 h. These results indicate the low toxicity of CDs, even after a long period of exposure at high concentrations. In 2021, R. Atchudan et al. [141] used *C. elegans* as an animal model for their toxicity evaluation of CQDs. As displayed in Figure 3, the CQDs synthesized from banana peel were shown to have low nematode toxicity, even at a high concentration of 200 ug/mL. These results can be explained by their utilizing non-toxic raw materials without adding any passivates or additives.

## 4. Theranostic Application of Natural Carbon Nanodots

Therefore, fluorescent materials are expected to be critical in biological applications. CDs have become the focus of attention as new nanomaterials due to their excellent light stability, fluorescence, good biocompatibility, and low toxicity. Surgery, radiotherapy, and chemotherapy are inevitable treatments for most cancer patients, but these processes have considerable side effects on the human body. However, fluorescent nanomaterials have the advantages of high fluorescence stability, low biotoxicity, and good biocompatibility. The most important thing is that they can be used to perform non-invasive targeted therapy on human lesions, exploiting their characteristics to avoid harm to the human body caused by the abovementioned treatments.

This paragraph covers the luminescence mechanism of CDs and their applications in biology, focusing on applying natural CDs in biological diagnosis and treatment. We discuss the combination of CDs with specific targeting molecules to form CD-based probes for detecting fluorescent signals. With the help of advanced optical imaging technology, real-time dynamic monitoring of molecules in cells and organisms can be carried out, and rapid immunofluorescence analysis of primary infectious disease sources can be carried out. They can provide new technologies and methods for disease occurrence, diagnosis, and treatment research [150].

### 4.1. Bioimaging

The discrete and diverse microstates of CDs lead to broad excitation and emission ranges [151,152]. CDs have many unique properties, and their excellent photostability can provide fluorescent information in a biological environment. The surface modification of functional groups can make CDs more helpful in applying biomarkers (Table 3). In this section, we compare bioimaging from cells to living animals, emphasizing the biological diversity to prove the generalized safety of CDs. In cell imaging applications, CDs are usually applied to HepG2 cells, HeLa cells, T24 cells, and so on [27,102,120,124]. In addition, normal cell lines have been used in trials, showing good cell compatibility [24,71]. Alam et al. treated HaCaT cells with cabbage-derived CDs and showed that, at 500 μg/mL of CDs, the cell viability was 100%. Furthermore, CDs’ tunable excitation and emission show promise for normal cell imaging under the irradiation of confocal fluorescence microscopy [71]. In a cell imaging experiment, Mehta et al. considered CDs originating from apple juice and three fungal (*M. tuberculosis*, *P. aeruginosa*, and *M. oryzae*) sources. Cell compatibility was shown by feeding more than 100 mg/mL of CDs. The germination of *M. oryzae* spores also strongly indicated how CDs are good biocompatible nanocomposites at high concentrations (i.e., 400 mg/mL). The fungi appeared red, green, and blue in confocal laser microscopic images [62].

Kasibabu et al. provided pictures of *Bacillus subtilis* and *Aspergillus aculeatus* after incubating with CDs derived from papaya juice for 1–6 h. The uptake ability was shown by observing well-dispersed CDs in the cytoplasm of the fungi [90]. In vivo tests are critical standards for investigating the potential and toxicity of CDs in animals. Atchudan et al. explored colorful nitrogen-doped CDs (NCDs) derived from gooseberry by hydrothermal methods in *C. elegans* imaging. *C. elegans* presented blue, green, and red in the whole body when excited under 400 nm, 470 nm, and 550 nm. The cell viability was over 97% after incubating *C. elegans* in NCDs for 24 h from 0 to 200 μg/mL (Figure 4a–f) [96]. Cong et al., using roast duck as the source and pyrolyzing at 200 °C, 250 °C, and 300 °C, synthesized single-fluorescence CDs. *C. elegans* were treated with 15 mg/mL of the CDs pyrolyzed at 300 °C (300-CDs) for 24 h. Compared with the wild-type group, the accumulation and uptake of 300-CDs made the intestine appear blue under UV light exposure (Figure 4g–j) [122]. The murine model has also been widely used for determining the efficiency of CDs.

Ding et al. subcutaneously injected 100 μL of red-emitting CDs (R-CDs) into nude mice. Strong fluorescence at 700 nm was detected under an excitation wavelength of 535 nm, indicating the excellent penetration ability from tissues to skin. Furthermore, the mice were still alive after 10 days (Figure 5a) [22]. Liu et al. investigated how the accumulation of carbonized polymer dots (CPDs) varied with time. Most CPDs remained in the lung and liver in the early period, with negligible dispersion to the brain and heart. The metabolism of CPDs was confirmed after 4 h, as their fluorescence decreased sharply (Figure 5b) [23]. Ding et al. compared subcutaneous and intravenous injections of near-infrared emissive CDs (NIR-CDs). Through the use of subcutaneous methods, NIR-CDs were distributed into mouse skin and tissues. As for intravenously injected nude mice, fluorescence was seen in the bladder, indicating NIR-CDs elimination via urine (Figure 5c) [26]. Liu et al. tracked CDs (Fn-CDs) in Kunming mice for different periods.

They concluded that, with the assistance of PL, Fn-CDs show strong fluorescence in the bladder and are eliminated after 7 h (Figure 5d) [28]. Therefore, CDs can perform well as imaging nanoparticles and be adapted to different cell lines and living animals. In addition, their multiple emission ranges, low toxicity, and small size confer their high potential in future clinical applications.

### 4.2. Sensors

Inorganic ions are critical for creatures, not only for enhancing the efficiency of catalytic reactions in bio-systems but also for maintaining our fundamental life functions. However, it is harmful to have too many metal cations, which are highly toxic to human beings. Therefore, developing sensors for inorganic ions is a simple beneficial method to collate the concentration and standard. Owing to the multiple PL of CDs, we can observe the intensity variation and chelation quenching effect at other peaks in the PL image. For natural CDs sensors, Fe^3+^, Hg^2+^, and Cu^2+^ are the most examined targets. Shen et al. assessed HepG2 and HeLa cell images after incubating with CDs and Fe^3+^.

According to the decreased blue fluorescence under 405 nm irradiated light, Fe^3+^ had a practical quenching effect on the CDs (Figure 6a–d) [33]. Hu et al. prepared double--emitted biomass nitrogen co-doped CDs (B-NCdots) for Cu^2+^ probing in T24 cells. Similarly, the quenching effect was still available for Cu^2+^, causing a decreased intensity of blue and green, as shown in Figure 6e,f [117]. Furthermore, bio-related molecules, including peptides and drug-containing cells, are even more crucial. Some researchers designed chemically sensitive CDs to assist in directly resolving the effects of molecules by monitoring the decrease or increase in fluorescence. Liang et al. added 0.5 mM and 1 mM of glutathione and 150 μg/mL rose-red fluorescence CDs (wCDs) in L929 cells, HeLa cells, and HepG2 cells. An intense quenching effect of glutathione was observed in L929 cells and HeLa cells. However, no apparent variation occurred in HepG2 cells, which implies a distinct response of wCDs to glutathione (GSH) in various cells (Figure 7) [24].

Wang et al. constructed a glutathione assay composed of eggshell-derived CDs and Cu^2+^. The authors quantify without other indicators by plotting the fluorescence ratio versus the glutathione concentration (Figure 8a) [107]. Wang et al. examined how the fluorescence intensity of Shiitake mushroom-derived CDs (MCDs) varied with pH in dexamethasone-induced HeLa cell apoptosis. At higher concentrations of dexamethasone, the fluorescence under excitation at 405 nm and 488 nm was stronger, indicating a positive relationship on MCDs with increasing intracellular acidification (Figure 8c–e) [66]. As for drug probing, Zhu et al. analyzed doxorubicin (DOX), an anthracycline-based anticancer medicine, by taking advantage of the PL of plum-based carbon quantum dots (PCQDs). The dual-emitted property at the wavelengths of 491 nm and 591 nm provided a ratiometric calibration curve as a function of the DOX concentration. They also confirmed the accuracy by analyzing urine and serum samples (Figure 8b) [27]. CDs are suitable for detecting different kinds of molecules and ions. The intensity changed at a single emission peak, but the amplitude ratio of two emission peaks is valid for sensing experiments. Due to their intrinsic properties, CDs show promise in the bio-sensing field and are applied to cancer therapy.

### 4.3. Antibacterial Activity

Bacteria are well-known as the origins of various diseases. Recently, super bacteria have appeared globally, which cause incurable illnesses due to the abuse of antibiotics. In addition, people have come to pay more attention to the side effects of antibiotics and wish to avoid unexpected risks. Nanomedicines, especially CDs, have been taken into consideration as substitute methods. *E. coli* and *S. aureus* are the most common types of bacteria for investigating how nanomedicines or antibiotics work to induce apoptosis in bacteria. Wang et al. used CDs (ACDs) derived from *Artemisia argyi* leaves to treat *E. coli* and *S. aureus* cultures. According to the SEM images (Figure 9a–h), it can be seen that the cell walls of *E. coli* were destroyed; however, there was no distinct difference between treated and untreated *S. aureus*. This means that ACDs are selective to Gram-negative bacteria due to the structural properties of their cell walls [123]. Sun et al. synthesized chlorhexidine gluconate CDs from large to small (l-CGCDs, m-CGCD, and s-CGCDs) to determine the relationship between size and antibacterial activity. From the SEM imagery (Figure 9i), it can be seen that the rigidity of cell walls was the strongest in the control group and decreased from l- to s-CGCDs groups. These results revealed that CGCDs lead to frustration in the walls and membranes of *E. coli* and *S. aureus*. Bacterial death can be controlled by tuning the size of the CGCDs [76]. Ma et al. tested three kinds of CDs, including osmanthus leaves-derived CDs (OCDs), milk vetch-derived CDs (MCDs), and tea leaves-derived CDs (TCDs). In Figure 10, 80% of *E. coli* and *S. aureus* were killed by OCDs at a 1 mg/mL concentration, while 70% of bacteria survived in the MCDs group. In addition, *E. coli* had stronger resistivity than *S. aureus* among these CDs. CDs are internalized into bacteria. The outer surface of bacteria is attached to CDs leading to indirect proliferating inhibition [153,154,155]. These results prove the natural sources are essential for the synthesis of CDs [79]. As mentioned above, the tunability of raw material and diameters primarily affect the antibacterial efficiency and selectivity of the resultant CDs. Treating the patient’s wounds after surgery with CDs with editable properties can tremendously decrease the associated risks.

### 4.4. Anticancer Activity

At present, cancers are prevalent within all age ranges. Cancers may be fatal due to unexpected syndromes as well as the disorder of living functions. Furthermore, conventional cancer therapies are long-term processes. Surgeries are straightforward methods, but recovery typically poses a challenge for patients. Even though chemotherapy seems safer, the currently used drugs lack selectivity and affinities to specific tumors. Some targeted therapies have been developed in recent years. However, they are expensive and only valid for certain types of cancers. CDs can provide great theranostic nanomedicines in cancer treatments. Scientists have attempted to eliminate cancer cells through photothermal therapy (PTT) [81,82] and photodynamic therapy (PDT) [83,87,129] to fulfill tumor targeting. Li et al. tested NIR-II emitted (900–1200 nm) CDs (CDs), adapted for 808 nm laser photothermal therapy. According to the in vivo test (Figure 11), the temperature increased to 50 °C in the intratumoral environment after intravenous injection. Tumor inhibition and volume shrinkage were observed within 6 days, compared with the PBS group. No detectable damage to tissues or weight loss after the treatment confirmed the high biocompatibility of the CDs [156]. Jia et al. prepared red-light absorbing (610 nm) CDs from *Hypocrella Bambusa* (HBCUs). They found that HBCDs highly generate ^1^O_2_ under 635 laser irradiation. The reactive radicals induced apoptosis of cancer cells, which is helpful in the hypoxia tumor environment.

As shown in an in vivo experiment (Figure 12), due to the synergistic effect of PDT and PTT, the temperature at the tumor site increased to 56.4 °C in 10 min. Secondly, a good tumor inhibition effect was found after 14 days of therapy, even though the tumor could not be depleted thoroughly. No harmful phenomena were observed in other organs, indicating the safety of the treatment [82]. Xue et al. conducted modification with polyethylene glycol diamine (H_2_N-PEG-NH_2_), chlorin e6 (Ce6), and transferrin (Tf) on natural biomass CDs (NBCDs) to increase the targeting efficacy.

The resulting products, NBCD-PEG-Ce6-Tf, were shown to remain within the tumor environment for 120 h using a real-time NIR fluorescence image (Figure 13a). The mice were irradiated daily under a 650 nm laser to generate ^1^O_2_. During the 21-day process, tumor growth was stopped, and the tumors were ablated, indicating no conflict between the NBCDs and modifications (Figure 13b,c) [126]. Li et al. synthesized reactive oxygen species (ROS)-generating CDs from ginger. The CDs were harvested from HepG2 tumor inoculating mice; next, the tumor regressions were observed in the C-dot (440 μg) treated group; the tumor growth was prominently delayed, which attained only 3.7 ± 0.2 mg. In contrast, tumors in the PBS group grew up to 104 mg [84]. Boobalan et al. added 30 μg/mL of CDs into *P. aeruginosa*. They observed the destruction of cell walls due to ROS attack, in agreement with the results of ROS fluorescence detection using a fluorogenic dye, 2′,7′-dichlorofluorescein diacetate (DCFDA) (Figure 14a–c). MDA-MB-231 breast cancer cells were treated with CDs (3.34 μg/mL). Cell apoptosis staining, acridine orange and ethidium bromide (AO/EtBr), and nuclear staining (Hoechst 33342) were applied to the cells. The presence of orange colors and blue dots indicate cell fragmentation due to CDs (Figure 14d–g) [80]. CDs have a powerful potential in the anticancer field. Their flexibility is because the CDs are modified with various molecules, which can improve the uptake by tumor cells and increase the tumor-killing ability of the nanohybrids.

## 5. Discussion and Conclusions

Carbon nanomaterials have been widely used in various scientific, engineering, and commercial fields, due to their high catalytic activity and good stability. Among them, the new “zero-dimensional” carbon nanomaterials, CDs, have unique optical properties, such as stable fluorescence signals, no light scintillation, adjustable excitation and emission wavelength, and low biological toxicity and biocompatibility. These advantages gradually led to the popularity of researching carbon nanomaterials, widely used in bioimaging, natural cell labeling, sensors, photocatalysis, solar cells, and light-emitting elements. This article mainly reviewed the different synthesis methods of CDs (including top-down and bottom-up methods) and their applications. Their luminescence properties can be adjusted through surface modification. They have been applied in many fields and have great potential. The function of CDs can also be modified by using various surface functional groups, allowing them to act, for example, as detectors and cleaners for different heavy metal ions or by doping with other ions. By controlling the surface light energy groups, they can be better used in the required fields.

Green chemistry is a discipline that has gradually received attention in recent years. The core concept focuses on the development of environmentally friendly chemical technologies. At the technical level, chemical technologies and methods are applied to reduce or eliminate the use and generation of hazardous substances in chemical synthesis and analysis, and recovery and reuse technologies are combined with increasing energy and material use efficiency. Green chemistry and nanotechnology have become emerging technology research and development directions in recent years. With the deepening of the concept of sustainability, combining the advantages of the two and accelerating the expansion of their research and development applications has become a top priority. Laboratories are committed to determining the complementary relationships between green chemistry practices and nanotechnology and applying them to materials development, chemical analysis, energy, environmental, and other related fields. The preparation of CDs well embodies the concept of green chemistry. Cheap, environmentally friendly carbon source precursors and natural renewable raw materials as carbon sources for preparation, such as eggs, grass, leaves, silk, coffee grounds, beer, and other materials, have become carbon sources for the synthesis of CDs.

Excellent performance and a unique structure provide natural CDs unlimited charm and various changes. Natural CDs, combined with biological and pharmaceutical molecules of interest through surface modification, seem to be an emerging platform for imaging probes that are both diagnostic and therapeutic. The next generation of nano-molecular probes integrates a variety of fluorescent dyes, drugs, and multifunctional nanomaterials into a single nanoprobe, providing superior signal contrast, controllable transmission, and targeted drug delivery capabilities. However, before this kind of multifunctional imaging probe was used in diagnosis and treatment, there were still many challenges, such as long-term safety, risk-benefit, biocompatibility, and biodistribution, to be evaluated. In the future, we need to solve several critical scientific problems in the research of natural CDs. First of all, the uncertain chemical groups on the surface indicate that natural-synthetic CDs are a kind of unsure material. It means the method of natural mass production of high-quality CDs is still a big challenge. Secondly, due to the different sources of naturally transformed CDs, the luminescence centers of CDs are also dissimilar. Finding a suitable luminescence position is also essential to research content. Third, categorizing different natural CDs and conducting a systematic comparative analysis will be beneficial research methods. Nanotoxicology is the emerging study of potential adverse effects derived from the interaction between nanomaterials and biological systems, and it is bound to become more critical. To further clarify its physical toxicity and adjust the size and structure accordingly, it will significantly improve CDs’ application performance while expanding a more comprehensive range of applications. The scale and complexity of biomedical issues have always been an enormous challenge for researchers. It is more necessary to conduct cross-disciplinary research in chemistry, physics, materials, biomedical engineering, toxicology, public health, and clinical medicine. As more research on natural CDs continues to develop, the topic will achieve breakthroughs and progress in a short period.

## Figures and Tables

**Figure 1 pharmaceutics-13-01874-f001:**
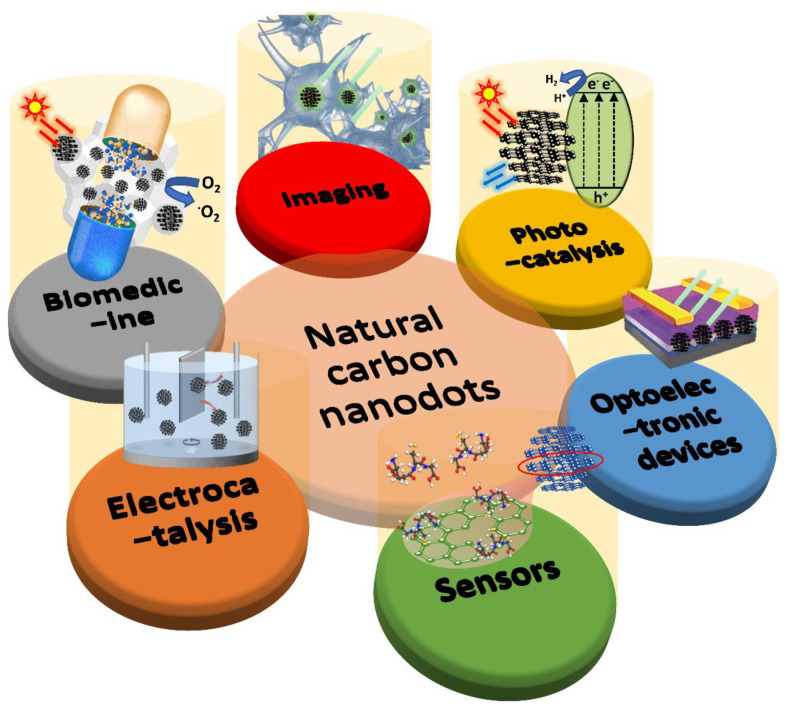
The main application research directions of CDs. This review focuses on summarizing the biological applications of CDs, such as cell imaging, photocatalysis, optoelectronic devices, molecules sensors, electrocatalysis, and biomedicines, comprehensively. Finally, current challenges, research emphasis, and prospects of this field are also discussed.

**Figure 2 pharmaceutics-13-01874-f002:**
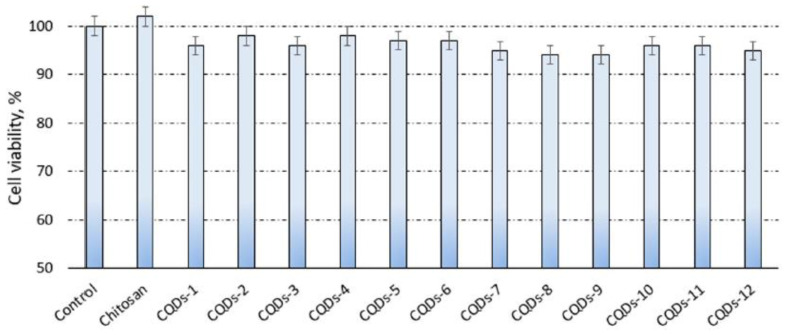
TXT assay study on human dermal fibroblasts. Adapted from [129], MDPI, 2019.

**Figure 3 pharmaceutics-13-01874-f003:**
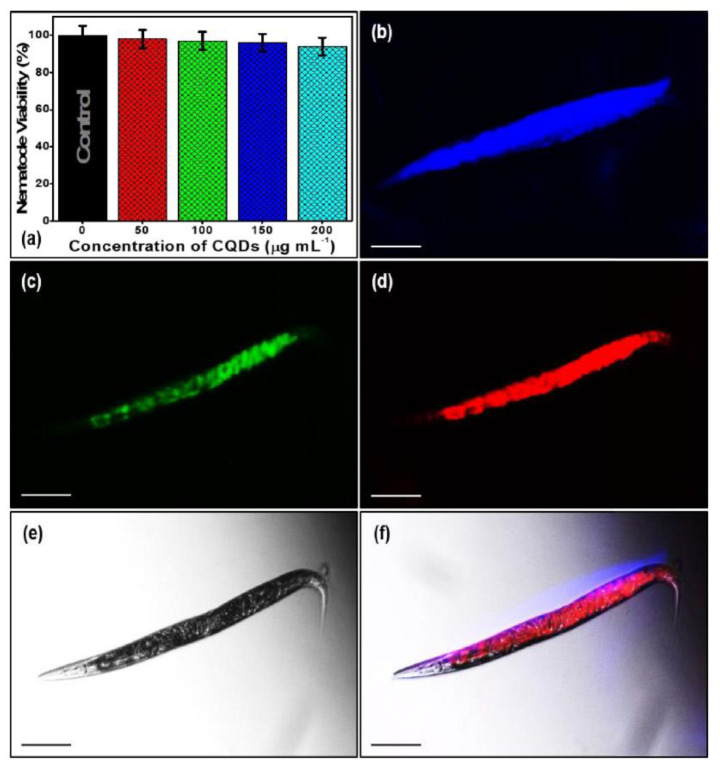
(**a**) Toxicity assay of nematode incubation under different concentrations of synthesized CQDs. Multicolor imaging of in vivo model nematode incubated with CQDs (100 μg/mL) under the excitation wavelengths of (**b**) 400 nm, (**c**) 470 nm, (**d**) 550 nm, (**e**) bright-field (BF), and (**f**) merge (overlap). Live nematodes were immobilized using 0.05% sodium azide (NaN_3_) for imaging under fluorescence filters. Adapted with permission from [141], Elsevier, 2021.

**Figure 4 pharmaceutics-13-01874-f004:**
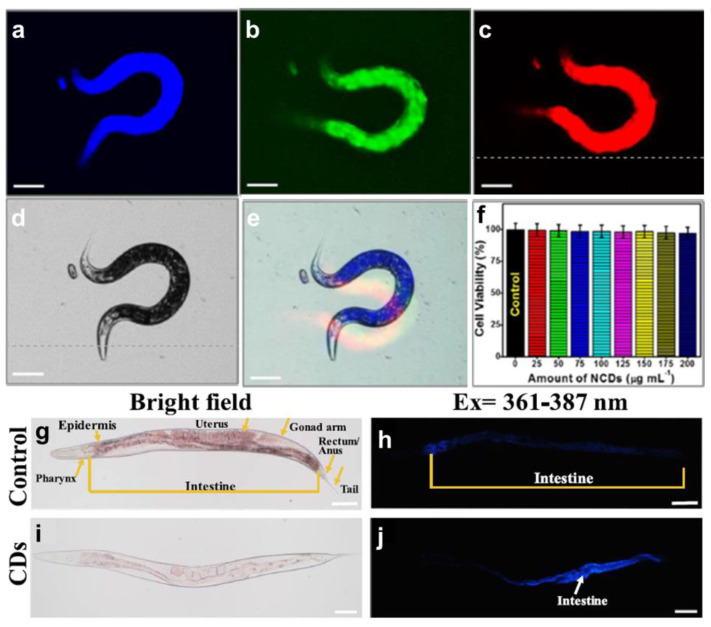
*C. elegans* confocal imaging excited under the wavelengths of (**a**) 400 nm, (**b**) 470 nm, and (**c**) 550 nm; as well as (**d**) bight field; (**e**) the merged image; and (**f**) cell viability test under different concentrations of NCDs, adapted with permission from [96], ACS Publications, 2018. Bright-field (**g**) wild-type *C. elegans* imaging and (**i**) 300-CDs treated *C. elegans* imaging; UV-exposed (**h**) wild-type *C. elegans* imaging and (**j**) 300-CDs treated *C. elegans* imaging, adapted with permission from [122], Elsevier, 2019.

**Figure 5 pharmaceutics-13-01874-f005:**
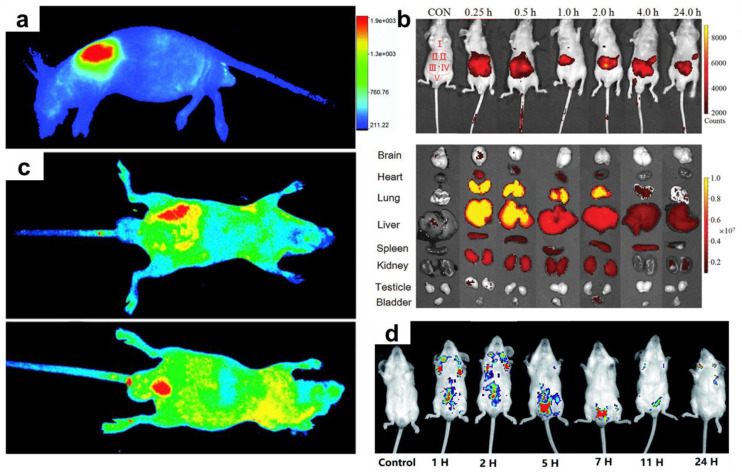
In vivo images and accumulation of CDs: (**a**) in vivo images of R-CDs, adapted with permission from [22], Royal Society of Chemistry, 2017; (**b**) metabolism of CPDs, adapted with permission from [23], Wiley, 2020; (**c**) subcutaneous and intravenous injection of NIR-CDs, adapted with permission from [26], Elsevier, 2019; and (**d**) metabolism of Fn-CDs, adapted with permission from [28], Royal Society of Chemistry, 2021.

**Figure 6 pharmaceutics-13-01874-f006:**
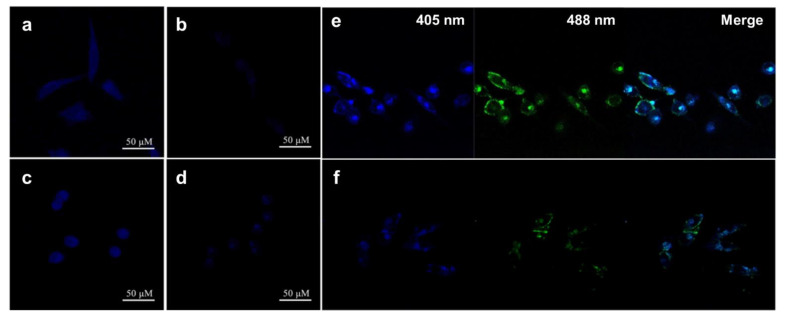
Confocal images collected at 405 nm of (**a**) HeLa cell; (**c**) HepG2 cell incubated with CDs; adding Fe^3+^ and CDs to (**b**) HeLa cell; and (**d**) HepG2 cell, adapted with permission from [33], Elsevier, 2017. Confocal laser images of T24 cells excited at 405 nm and 488 nm, (**e**) treated with B-NCdots and (**f**) treated with Cu^2+^ and B-NCdots, adapted with permission from [117], Springer Nature, 2019.

**Figure 7 pharmaceutics-13-01874-f007:**
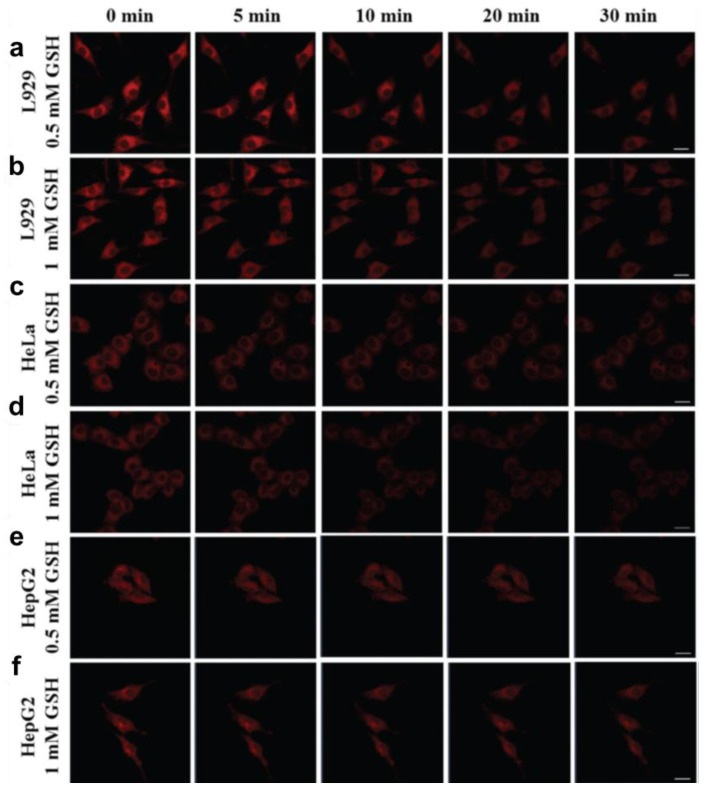
Confocal laser images of: (**a**) L929 cells, (**c**) HeLa cells, and (**e**) HepG2 cells in 0.5 mM GSH and 150 μg/mL wCDs; and (**b**) L929 cells, (**d**) HeLa cells, (**f**) and HepG2 cells in 1 mM GSH and 150 μg/mL wCDs, adapted with permission from [24], Royal Society of Chemistry, 2021.

**Figure 8 pharmaceutics-13-01874-f008:**
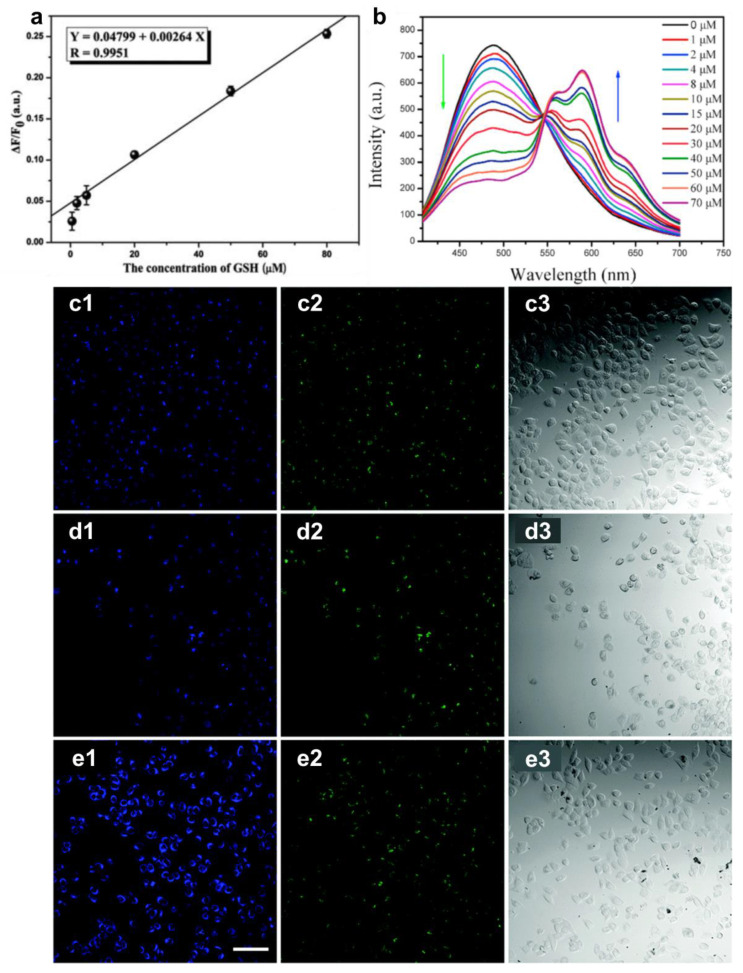
(**a**) Linear calibration of GSH probing, adapted with permission from [107], Royal Society of Chemistry, 2012. (**b**) The fluorescence spectra of PCQDs with varying DOX concentrations, adapted with permission from [27], Elsevier, 2021. (**c**) Confocal images of HeLa cells treated with MCDs, (**d**) adding 10 μM dexamethasone or (**e**) 100 μM dexamethasone, at excitation wavelengths of (**c1**,**d1**,**e1**) 405 nm, (**c2**,**d2**,**e2**) 488 nm, and (**c3**,**d3**,**e3**) bright-field, adapted with permission from [66], Royal Society of Chemistry, 2016.

**Figure 9 pharmaceutics-13-01874-f009:**
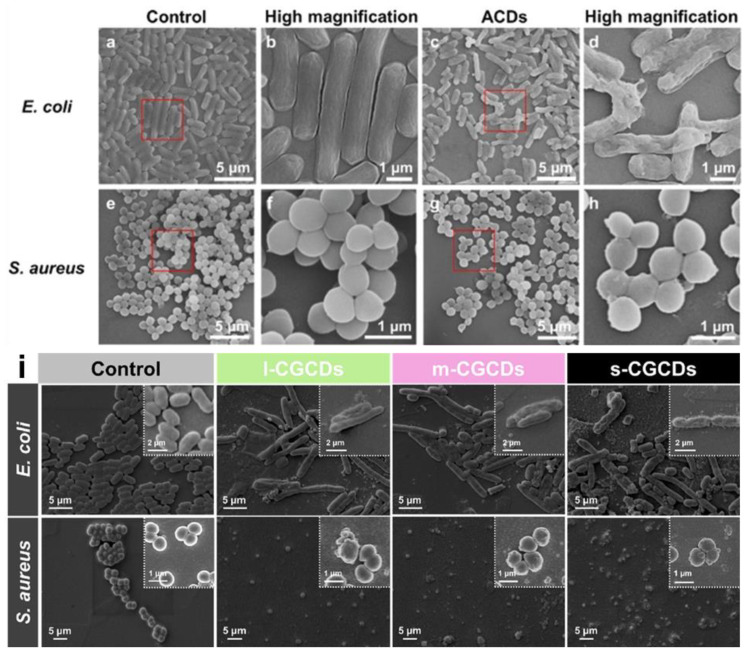
SEM images without ACDs of (**a**) *E. coli*, (**e**) *S. aureus*, and ACDs-treated (**c**) *E. coli*, and (**g**) *S. aureus*. Magnified SEM images in the red square (**b**) *E. coli*, (**f**) *S. aureus*; and ACDs-treated (**d**) *E. coli*, and (**h**) *S. aureus*, adapted with permission from [123], Royal Society of Chemistry, 2020. SEM images of (**i**) *E. coli* and *S. aureus* untreated and treated with 75 μg/mL and 50 μg/mL of s-CGCDs, m-CGCDs, and l-CGCDs in Luria–Bertani broth medium for 6 h, adapted with permission from [76], Elsevier, 2021.

**Figure 10 pharmaceutics-13-01874-f010:**
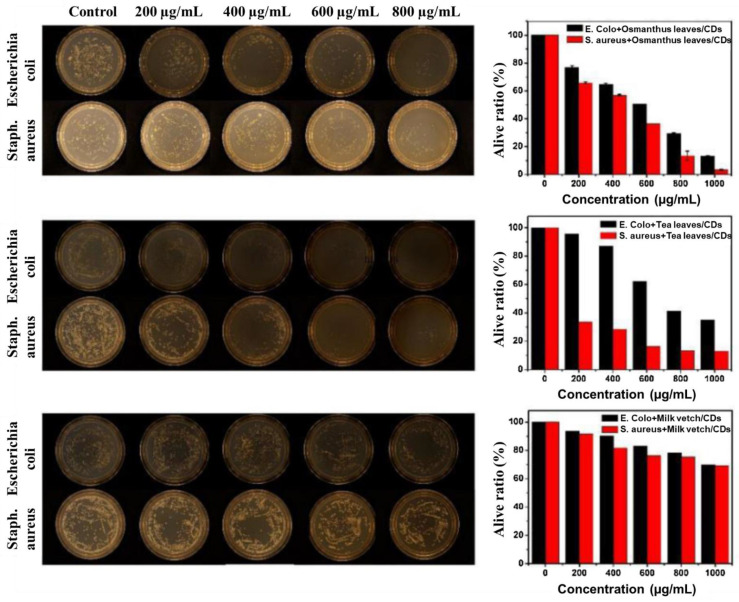
Graphs of *E. coli* and *S. aureus* incubated with varying concentrations of OCDs, TCDs, and MCDs for 24 h; alive ratio of *E. coli* and *S. aureus* calculated using UV–Vis spectroscopy methods, adapted with permission from [79], Elsevier, 2020.

**Figure 11 pharmaceutics-13-01874-f011:**
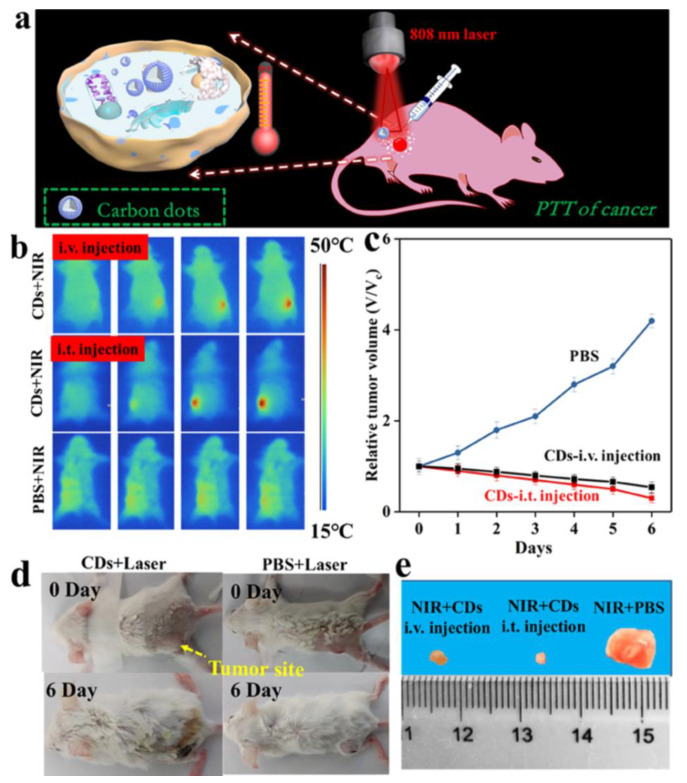
(**a**) Schematic diagram of photothermal therapy of cancer in vivo; (**b**) infrared thermal images of tumor-bearing mice with intravenous (i.v.) or intratumoral (i.t.) injection of CDs (50 μL, 20 mg/mL), and PBS (50 μL); (**c**) tumor volume after treatment; and (**d**,**e**) images of tumor-bearing mice and harvested tumors after 6 days, adapted with permission from [156], ACS publications, 2019.

**Figure 12 pharmaceutics-13-01874-f012:**
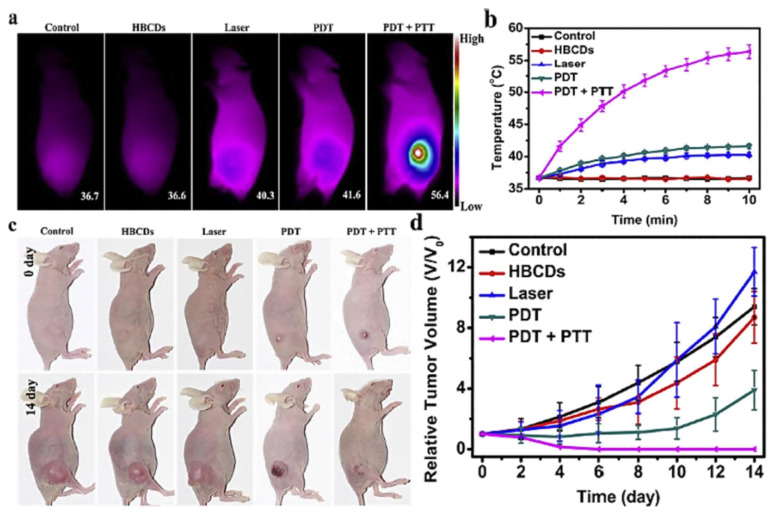
(**a**) IR thermal images post-injection; (**b**) temperature increase trends; (**c**) images of mice with different therapeutic methods; and (**d**) growth of the tumor during 14 days, adapted with permission from [82], Elsevier, 2018.

**Figure 13 pharmaceutics-13-01874-f013:**
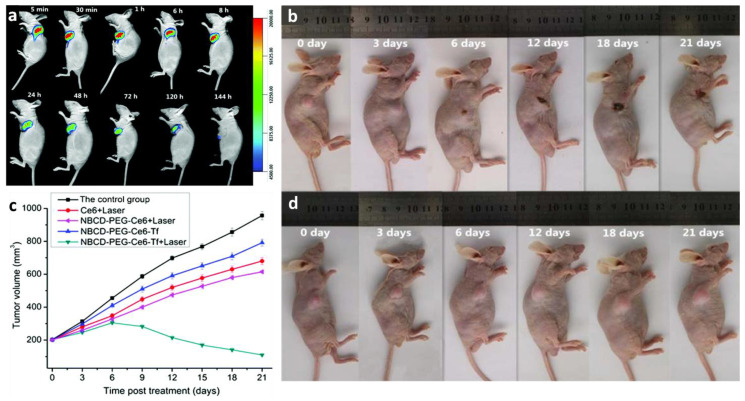
(**a**) Real-time NIR fluorescence image of mice tumors under NBCD-PEG-Ce6-Tf treatment at different time points; tumor propagating images for (**b**) NBCD-PEG-Ce6-Tf + laser-treated and (**d**) control groups; (**c**) time-dependent tumor growth curves after other treatments, adapted with permission from [126], Royal Society of Chemistry, 2018.

**Figure 14 pharmaceutics-13-01874-f014:**
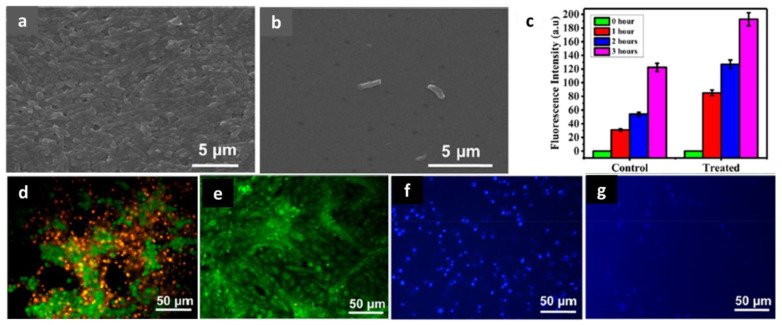
SEM images of (**a**) untreated and (**b**) treated *P. aeruginosa* with 30 μg/mL of CDs; (**c**) DCFDA probing of ROS generation; fluorescent microscopic images of MDA-MB-231 cells (**d**,**f**) CDs (3.34 μg/mL) treated and (**e**,**g**) untreated with nuclear staining assays (AO and Hoechst 33342), adapted with permission from [80], ACS publications, 2020.

**Table 1 pharmaceutics-13-01874-t001:** Natural CDs and their applications.

Source	Methods	Applied Ex./Emit. (nm)	QY(%)	Application	Applied Concentration	Ref.
Lemon juice	Solvothermal	440–600/575–650	28%	In vivo bioimaging	100 μL (20 μg/mL)	[22]
Taxus leaf	Solvothermal	380–580/673	59%	In vivo bioimaging	20 mg/kg	[23]
*Wedelia trilobata*	Solvothermal	370–470/483–520, 654	H_2_O: 10.52% EtOH: 18.16%	Glutathione sensing and cell imaging	50–400 μg/mL	[24]
Leek	Solvothermal	390–450; 325–385/440–500, 676; 440	DCD: 1.7% SCD: 1.14%	Cell imaging, Cu^2+^ and tetracycline sensing	0.5 mg/ mL	[25]
Lemon juice	Solvothermal	237, 279, 570–670/704	31%	In vivo bioimaging	50 μL (30 μg/mL)	[26]
Plum	Solvothermal	328–418/450–550	0.54%	Doxorubicin sensing	200 μL (1.0 mg/mL)	[27]
*F. nucleatum*	Hydrothermal	300–400/450–470	9.9%	In vivo bioimaging and Fe^3+^ sensing	10 μL/ g (0.7 mg/mL)	[28]
Green pepper	Hydrothermal	310–380/400–460	8.7%	Fe^3+^ sensing and cell imaging	50 mg/mL	[29]
Papaya	Hydrothermal	350–490/445–530	H_2_O: 18.98% EtOH: 18.39%	Fe^3+^ sensing and cell imaging	100, 175 μL (0.94 mg/mL) 100, 175 μL (1.17 mg/mL)	[30]
*P. avium*	Hydrothermal	280–360/411–430	13%	Fe^3+^ sensing and cell imaging	0–40 μL	[31]
Honey	Hydrothermal	320–410/410–475	19.8%	Fe^3+^ sensing and cell imaging	40 μL (1 mg/mL)	[32]
Sweet potato	Hydrothermal	300–410/406–486	8.64%	Fe^3+^ sensing and cell imaging	0–100 μg/mL	[33]
Black tea	Hydrothermal	290–420/398–490	n/a	Fe^3+^ sensing	990 μL (8 μg/mL)	[34]
Fish-scale	Hydrothermal	220–390/425–455	6.9%	Fe^3+^ sensing	5 mg/mL	[35]
Kiwi	Hydrothermal	300–450/432–500	14%/19%	Fe^3+^ sensing	0.5 mL	[36]
Goose feather	Hydrothermal	300–500/410–560	17.1%	Fe^3+^ sensing	1 mL	[37]
Cranberry	Hydrothermal	300–500/410–540	10.85%	Fe^3+^ sensing	n/a	[38]
Potato	Hydrothermal	323/405	15%	Fe^3+^ sensing	n/a	[39]
*Boswellia ovalifoliolata* bark	Hydrothermal	275–440/400–535	10.2%	Fe^3+^ sensing	20 μg/mL	[40]
Rosin	Hydrothermal	290–370/425–475	1.22%	Fe^3+^ sensing and cell imaging	1.25–160 μg/mL	[41]
Coriander leaf	Hydrothermal	320–480/400–510	6.48%	Fe^3+^ sensing, cell imaging, and antioxidant	0–1.0 mg/mL	[42]
Mint leaf	Hydrothermal	330–420/410–500	7.64%	Fe^3+^ and ascorbic acid sensing	n/a	[43]
Leek	Hydrothermal	300–460/449–534	n/a	DDVP sensing and cell imaging	0–300 mg/mL	[44]
Peach gum	Hydrothermal	330–450/327–505	28.46%	Au^3+^ sensing and cell imaging	0.5 mL (20 mg/mL)	[45]
Tomato	Hydrothermal	367/440	n/a	Carcinoembryonic antigen and aptamer sensing	1 μg/mL	[46]
Bean pod and onion	Hydrothermal	310–380/410–450	5.55%	Ag^1+^ sensing and cell imaging	200 μg/mL	[47]
*D. Salina*	Hydrothermal	310–400/400–475	8%	Hg^2+^ and Cr^4+^ sensing and cell imaging	0–75 μg/mL	[48]
Chinese yam	Hydrothermal	280–440/400–525	9.3%	6-mercaptopurine and Hg^2+^ sensing	n/a	[49]
Pomelo peel	Hydrothermal	365/444	6.9%	Hg^2+^ sensing	n/a	[50]
Strawberry	Hydrothermal	344–440/427–500	6.3%	Hg^2+^ sensing	75 μL	[51]
Cucumber	Hydrothermal	418–518/514–571	3.25%	Hg^2+^ sensing	n/a	[52]
Highland barley	Hydrothermal	340–480/450–525	14.4%	Hg^2+^ sensing	0.05 mg/mL	[53]
Lemon peel	Hydrothermal	300–540/441–605	14%	Cr^6+^ sensing	n/a	[54]
*Elaeagnus angustifolia*	Hydrothermal	310–410/290–450	16.8%	Tartrazine sensing	n/a	[55]
Aloe	Hydrothermal	370–480/443–525	10.37%	Tartrazine sensing	n/a	[56]
Coconut water	Hydrothermal	340–450/430–500	2.8%	Thiamine sensing and cell imaging	n/a	[57]
Lentil	Hydrothermal	310–390/400–460	10%	Thioridazine hydrochloride sensing	200 μL	[58]
Pomegranate juice	Hydrothermal	280–350/350–600	4.8%	Cephalexin sensing	30 μL (1.0 mg/mL)	[59]
Bamboo leaf	Hydrothermal	365–525/440–540	7.1%	Cu^2+^ sensing	n/a	[60]
Pipe tobacco	Hydrothermal	310–430/425–515	3.2%	Cu^2+^ sensing	n/a	[61]
Apple juice	Hydrothermal	300–540/465–565	4.27%	Cell imaging	10 μg/mL	[62]
*Hylocereus undatus*	Hydrothermal	275–380/400–450	n/a	Cell imaging	0–50 μL/mL	[63]
*Saccharum officinarum*	Hydrothermal	300–540/450–550	5.76%	Cell imaging	0–400 mg/mL	[64]
Linseed	Hydrothermal	350–450/503	61%	Cell imaging	0.04 mg/mL	[65]
Shiitake mushroom	Hydrothermal	330–450/410–500	5.5%	Cell imaging and pH sensing	2 mg/mL	[66]
Citrus	Hydrothermal	360–500/460–554	1.1%	Cell imaging	30 μL (1.0 mg/mL)	[67]
Carrot	Hydrothermal	360–520/442–565	5.16%	Cell imaging	700 μg/mL	[68]
Dwarf banana	Hydrothermal	310–460/395–505	23%	Cell imaging	0–200 μg/mL	[69]
Bagasse	Hydrothermal	330–510/450–550	12.3%	Cell imaging and biolabeling	100 μg/mL	[70]
Cabbage	Hydrothermal	276, 320/432–584	16.5%	Cell imaging	100 μL (20–1000 μg/mL)	[71]
Alkali lignin	Hydrothermal	280–450/410–510	21%	Cell imaging	0–100 μg/mL	[72]
Shrimp	Hydrothermal	360–530/430–550	54%	Cell imaging and drug delivery	10–500 μg/mL	[73]
Wheat bran	Hydrothermal	360–540/460–600	33.23%	Cell imaging and drug delivery	2 mg/mL	[74]
Milk	Hydrothermal	360/450	n/a	Cell imaging and anticancer drug delivery	100–600 μg/mL	[75]
Chlorhexidine gluconate	Hydrothermal	360–560/480–600	s-CGCDs: 2.6% m-CGCDs: 11.3% l-CGCDs: 8.0%	Antibacterial and cell imaging	0–150 μg/mL	[76]
Turmeric leaf	Hydrothermal	310–470/429–520	n/a	Anti-bacterial	0–1.0 mg/mL	[77]
Rosemary	Hydrothermal	332–422/424–500	n/a	Anti-bacterial	12 μg/mL	[78]
Osmanthus leaf, tea leaf, and milk vetch	Hydrothermal	450/530	n/a	Antibacterial and cell imaging	0–1000 μg/mL	[79]
Mushroom	Hydrothermal	300–500/372–545	n/a	Anti-bacterial, anti-cancer, and Pb^2+^ sensing	0–25 μg/mL	[80]
Watermelon	Hydrothermal	808/900–1200	0.4%	Photothermal therapy and cell imaging	0–20 mg/mL	[81]
*Hypocrella Bambusa*	Hydrothermal	540–590/600–650	n/a	Photodynamic and photothermal therapy	0–200 μg/mL	[82]
*Camellia japonica*	Hydrothermal	360/400–700	n/a	Photodynamic and photothermal therapy	45 μg/mL	[83]
Ginger	Hydrothermal	325–445/400–500	13.4%	Cancer inhibition and cell imaging	440 μg	[84]
Garlic	Hydrothermal	320–580/380–600	17.5%	Cell imaging	0–1 mg/mL	[85]
Starch	Hydrothermal	340–500/452–545	21.7%	Cell imaging	0.078–1.250 mg/mL	[86]
Orange juice	Hydrothermal	360–450/441–510	26%	Cell imaging	0–200 μg/mL	[87]
Bee pollen	Hydrothermal	340–450/425–505	c-CDs: 8.9% l-CDs: 6.1%	Cell imaging	0.5 mg/mL	[88]
Gelatin	Hydrothermal	300–500/430–580	31.6%	Cell imaging	5.o mL (0.8 mg/mL)	[89]
Papaya	Hydrothermal	300–500/450–550	7.0%	Cell imaging	16.2–500 μg/mL	[90]
Oatmeal	Hydrothermal	280–460/410–504	37.4%	Cell imaging	1 mg/mL	[91]
Egg white	Hydrothermal	290–450/415–540	61%	Cell imaging	0.04 mg/mL	[92]
Corn flour	Hydrothermal	320–500/401–553	7.7%	Cell imaging and Cu^2+^ sensing	0–640 μg/mL	[93]
Humic acid	Hydrothermal	320–520/440–540	5.7%	Cell imaging	0.2 mg/mL	[94]
Durian	Hydrothermal	400–560/605	79%	Cell imaging	0–500 μg/mL	[95]
Gooseberry	Hydrothermal	300–500/406–545	13.5%	*C. elegans* bioimaging	50 μg/mL	[96]
Rice husk	Hydrothermal	310–340/360–440	8.1%	Cell imaging	50 μg/mL	[97]
Ayurvedic	Chemical ablation	430/518	n/a	Cell imaging and phototherapy	0.5 mg/mL	[98]
Coffee bean shell	Chemical ablation	280–520/368–557	n/a	In vivo bioimaging and antioxidant	0–400 μg/mL	[99]
Muskmelon	Chemical ablation	342, 415, 425/432, 515, 554	7.07%/26.9%/14.3%	Hg^2+^ sensing and Cell imaging	0.25–1.00 mg/mL	[100]
Cow manure	Chemical ablation	320–450/400–530	0.65	Cell imaging	2.5 mg/mL	[101]
Food waste	Ultrasound irradiation	330–405/400–470	2.85%	Cell imaging	0–4 mg/mL	[102]
*Citrus limone* juice	Ultrasound irradiation	230–450/325–538	12.1%/15%	Cell imaging	2–100 mM	[103]
Crab shell	Ultrasound irradiation	330–390/410–450	14.5%	Cell imaging	0–1000 μg/mL	[104]
Silkworm	Microwave	300–400/350–550	46%	Cell imaging	0–15 mg/mL	[105]
Algal bloom	Microwave	300–500/400–550	13%	Cell imaging	10–1000 μg/mL	[106]
Eggshell	Microwave	275/450	14%	Glutathione sensing	n/a	[107]
Flour	Microwave	360–500/438–550	5.4%	Hg^2+^ sensing	4 μg/mL	[108]
Protein	Microwave	300–420/380–480	14%	Ag^+^ sensing	n/a	[109]
Rose flower	Microwave	330–410/390–435	13.45%	Tetracycline sensing	n/a	[110]
Onion peel	Microwave	300–470/520	n/a	Skin wound healing	1.5 mg/mL	[111]
Lychee	Pyrolysis	365/440	10.6%	Cell imaging	0–1000 μg/mL	[112]
Coffee	Pyrolysis	350–500/400–600	3.8%	Cell imaging	1.2 mg/mL	[113]
Urine	Pyrolysis	275–625/450–650	14%	Cell imaging	0.05–1.5 mg/mL	[114]
Watermelon peel	Pyrolysis	310–550/490–580	7.1%	Cell imaging	n/a	[115]
Konjac flour	Pyrolysis	400–700/575–760	13%	Fe^3+^ and L-lysine sensing and cell imaging	200 μg/mL	[116]
Soybean and broccoli	Pyrolysis	300–460/425–500	12.8%	Cu^2+^ sensing and cell imaging	0–300 μg/mL	[117]
*Borassus flabellifer*	Pyrolysis	300–400/350–403	11.73%/13.97%/10.83%	Fe^3+^ sensing	n/a	[118]
Peanut shell	Pyrolysis	262–402/413–500	10.58%	Cu^2+^sensing	n/a	[119]
Assam tea	Pyrolysis	340/446	n/a	Dopamine and ascorbic acid sensing	n/a	[120]
Peanut shell	Pyrolysis	320–480/441–524	9.91%	Cell imaging	0–1.2 mg/mL	[121]
Roast duck	Pyrolysis	300–400/400	10.53%/38.05%	*C. elegans* bioimaging	15 mg/mL	[122]
*Artemisia argyi* leaf	Pyrolysis	360–440/450–480	n/a	Antibacterial and cell imaging	0–150 μg/mL	[123]
Sugarcane bagasse	Pyrolysis	405/550	n/a	Drug delivery	n/a	[124]
Silkworm cocoon	Pyrolysis	378/459	6.32%	Anti-inflammatory	1.4 mg/mL;	[125]
Lychee exocarp	Pyrolysis	365/423	n/a	Drug delivery and cell imaging	0–15 μg/mL	[126]
Bamboo leaf	Pyrolysis	300–400/425–475	n/a	Cell imaging and anticancer drug delivery	0–400 μg/mL	[127]
Walnut shell	Pyrolysis	360–540/500–560	n/a	Cell imaging	100 μg/mL	[128]

**Table 3 pharmaceutics-13-01874-t003:** Modifications of CDs.

CDs Type	Modification	Goal	Ref.
Carbon quantum dots	Ethylene diamine	Nucleoli selection	[101]
Nitrogen-doped carbon dots	Folic acid	Cancer cell targeting	[104]
Carbon dots	Polyethylene glycol diamine; chlorin e6; transferrin	Photosensitizing and cancer cell targeting	
Carbon dots	4-carboxy-benzyl boronic acid	Tumor cell targeting	[127]

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
