# Peer review of "Natural Carbon Nanodots: Toxicity Assessment and Theranostic Biological Application"

_pharmaceutics, 2021, doi:10.3390/pharmaceutics13111874_

Round 1
Reviewer 1 Report
The present work offers a general overview of the synthesis of natural carbon dots and their potential use in different bioapplications. The review reads smoothly and clear and the authors highlight the main advantages of that kind of nanomaterials, including a high number of recent references. I list some minor concerns that need to be addressed,
1- Figure 1. To improve the reading in the figure, please use hyphens to separate the words inside each circle.
2- Line 59. The referenced work is not clear.. I understood that they compared the cytotoxicity of silicon, zinc oxides, CDs, single-walled carbon tubes, and carbon dioxide, and finally CDs showed the lowest toxicity, that’s right? Please, modify for a better understanding.
3- Line 365, the value 440 ug in brackets, what does it mean?
4- Quality of the Figure 5, Figure 7 top, Figure 8, Figure 9 needs to be improved.
Author Response
Response to reviewer 1:
Reviewer #1 (Reviewer Comments to the Author):
The present work offers a general overview of the synthesis of natural carbon dots and their potential use in different bioapplications. The review reads smoothly and clear and the authors highlight the main advantages of that kind of nanomaterials, including a high number of recent references. I list some minor concerns that need to be addressed,
Response: We thank the reviewer for your positive encouragement and kind suggestions. This review tries to integrate references to introduce nature carbon dots and put this nanoparticle forward to applying biological therapeutic and diagnosis in one system. Furthermore, we are also actively participating in related research to make this field more developmental.
1- Figure 1. To improve the reading in the figure, please use hyphens to separate the words inside each circle.
Response: Thank you for your kind suggestion. We have used hyphens to separate the words inside each circle. Please kindly check the revised Figure 1 as below.
2- Line 59. The referenced work is not clear.. I understood that they compared the cytotoxicity of silicon, zinc oxides, CDs, single-walled carbon tubes, and carbon dioxide, and finally CDs showed the lowest toxicity, that’s right? Please, modify for a better understanding.
Answer: We are thankful for the reviewer’s suggestion. We are sorry that the description in this sentence is not precise enough. The corrected sentence is as follows: “After the cytotoxicities of silicon and zinc oxide were determined, CDs showed the lowest toxicity compared with the materials mentioned above.” Please kindly check the revised Line 57-59 marked with a red highlight.
3- Line 365, the value 440 ug in brackets, what does it mean?
Answer: Thank you for your kind opinions. Here we revise the sentence to correct the meaning in it. The revised sentence is: “The CDs were harvested from HepG2 tumor inoculating mice; next, the tumor regressions were observed in the C-dot (440 μg) treated group; the tumor growth was prominently delayed, which attained only 3.7±0.2 mg.” Please kindly check the revised Line 400-403 marked with a red highlight.
4- Quality of the Figure 5, Figure 7 top, Figure 8, Figure 9 needs to be improved.
Answer: We appreciate the reviewer’s concerns. The resolution quality of those figures has been improved. Please kindly check the revised figures.

Reviewer 2 Report
The manuscript is interesting as it gives a rather full (not complete) overview of CDs prepared from various natural resources. However, it requires significant imrovement(s).
- The title “Natural carbon nanodots for theoretical evaluation and theranostic cancer application” is not compatible with the contents of the text. Theoretical evaluation is not discussed in the manuscript, while theranostic cancer application is not the only area covered in this review.
In the same context, a reorganization of the abstract is needed to depict the actual layout of the manuscript.
- The Introduction is referring to CDs in general (not just CDs derived from natural resources) and, therefore, more general references should be included, preferably reviews on this subject.
- The next section entitled “Natural Carbon Nanodots” should be devoted –as the title suggest- only to CDs derived from natural resources. Instead the first paragraph and most of the second paragraph and Figure 1 are dedicated to CDs in general, while in several instances they repeat themselves.
- Table 1 is informative and depicts the effort in compiling the relevant articles but it must complete. A number of papers on CDs derived from natural resources that are not mentioned in this table should be added (DOI: 10.1039/c3ra00088e; 10.1016/j.bios.2016.06.043; 10.1016/j.msec.2017.05.094).
Please also add for each entry the QY and concentration –or concentration range – employed for each specific application (when available). In addition, Table 2 is practically a part of Table 1 and it would be nice to combine both by adding in Table 1 another column with toxicity data (where available) or improve the Table 2 by adding all the entries of Table 1.
- In section 3 entitled “Theranostic Application(s) of Natural Carbon Nanodots” again parts of the text in the first paragraph are for CDs in general, while the second paragraph should be in the Introduction as it entails what the authors cover in this review.
In this section it is more appropriate to present first data on toxicity, then applications on cell imaging, followed by papers on animal imaging, and finally anticancer or antibacterial applications.
The description of CDs in many cases is not informative. The use of abbreviations without any explanation must be avoided. For instance, abbreviations such as R-CDs, CPDs, NIR-CDs, Fn-CDs etc. are meaningless- a description is needed for each one.
- In the section on “Antibacterial activity”, information must be included on the cell toxicity of these CDs at the concentrations that are effective against bacteria (if available). Is there any insight on what are the functional groups on their surface that lead to these properties?
- Several entries in Table 2 refer to CDs that are not derived from natural resources and therefore are not relevant.
- The Discussion and conclusion section is very small and should be improved.
Minor
Ref. 8 in line 64, ref. 126 in line 180 and ref. 127 in line 185 are irrelevant; Also Ref. 80 in Fig. 9 and in the relevant text is not the correct one. I suggest to thoroughly check all references.
Figure 1 mentioned in lines 224,228 is in fact Figure 2.
Please explain the term “structural frustration” in line 290.
“SEM imagery” should be replaced by “SEM images”
Author Response
Response to reviewer 2:
Reviewer #2 (Reviewer Comments to the Author):
Major
- The title “Natural carbon nanodots for theoretical evaluation and theranostic cancer application” is not compatible with the contents of the text. Theoretical evaluation is not discussed in the manuscript, while theranostic cancer application is not the only area covered in this review.
Answer: We are thankful for the reviewer’s suggestion. To make the title more consistent with all the content in the manuscript, we revised the new title: Natural carbon nanodots for its toxicity evaluation and theranostic biological application. Please kindly check the revised title for the manuscript.
- The Introduction is referring to CDs in general (not just CDs derived from natural resources) and, therefore, more general references should be included, preferably reviews on this subject.
Answer: We appreciate the reviewer’s concerns. We have modified the part of the introduction, deleted duplicate fragments, and updated the references. The revised paragraphs were highlighted with red markers. Please kindly check the revised section of the introduction with red highlights.
- The next section entitled “Natural Carbon Nanodots” should be devoted –as the title suggest- only to CDs derived from natural resources. Instead the first paragraph and most of the second paragraph and Figure 1 are dedicated to CDs in general, while in several instances they repeat themselves.
Answer: We thank the reviewer for the suggestions on separating the paragraphs to only CDs derived from natural resources. Here we intended to base on the concept of introduction to bring the natural carbon nanodots. Thus, we try to modify the section of the introduction to make the description fluently. We also gave some minor revisions to “Natural Carbon Nanodots.” Please kindly check the revised section of the introduction with red highlights.
- Table 1 is informative and depicts the effort in compiling the relevant articles but it must complete. A number of papers on CDs derived from natural resources that are not mentioned in this table should be added (DOI: 10.1039/c3ra00088e; 10.1016/j.bios.2016.06.043; 10.1016/j.msec.2017.05.094).
Answer: We thank the reviewer for the suggestions on the closely related references. Since we initially focused on the articles within five years, some references might be missing. The relative references have been added in the section of “Natural Carbon Nanodots.” and in Table 1. Please kindly check the revised references with a red-mark highlight.
References
- Biosens Bioelectron 2016, 86, 83-89.
- RSC Adv 2013, 3, 8286-8290.
- Mat Sci Eng C-Mater 2017, 79, 473-480.
- Please also add for each entry the QY and concentration –or concentration range – employed for each specific application (when available). In addition, Table 2 is practically a part of Table 1 and it would be nice to combine both by adding in Table 1 another column with toxicity data (where available) or improve the Table 2 by adding all the entries of Table 1.
Answer: Thank you for your kind opinions. Quantum yield and concentration have been added in Table 1 and Table 2. In addition, because the two subsections focus on different concepts, we revise the two tables to make the goals in the table not too complicated. Please kindly check the revised Table 1 and Table 2.
- In section 3 entitled “Theranostic Application(s) of Natural Carbon Nanodots” again parts of the text in the first paragraph are for CDs in general, while the second paragraph should be in the Introduction as it entails what the authors cover in this review.
Answer: Thank you for your kind concerns. Here we modify some sentences section of the “Theranostic Application of Natural Carbon Nanodots” to make the description fluently. Please kindly check the revised second paragraph in lines 218-220 with red highlights.
- In this section it is more appropriate to present first data on toxicity, then applications on cell imaging, followed by papers on animal imaging, and finally anticancer or antibacterial applications.
Answer: We appreciate the reviewer’s recommendation. Section 3 and section 4 have been changed to follow the orders of toxicity, cell imaging, animal imaging, and anticancer or antibacterial applications. Please kindly check the revised sections.
- The description of CDs in many cases is not informative. The use of abbreviations without any explanation must be avoided. For instance, abbreviations such as R-CDs, CPDs, NIR-CDs, Fn-CDs etc. are meaningless- a description is needed for each one.
Answer: We are thankful for the reviewer’s comments. We have added the abbreviations and listed those descriptions in the abbreviation list as below. Please kindly check the Abbreviations in the manuscript.
AO/EtBr |
Acridine orange and ethidium bromide |
B-NCdots |
Biomass nitrogen co-doped carbon dots |
c-CDs |
Camellia bee pollen carbon dots |
CDs |
Carbon dots |
CNDs |
Carbon Nanodots |
CPDs |
Carbonized polymer dots |
CQDs |
Carbon quantum dots |
CdSe |
Cadmium selenide |
CdS |
Cadmium sulfide |
ACDs |
CDs derived from Artemisia argyi leaves |
Ce6 |
Chlorin e6 |
l-CGCDs m-CGCDs s-CGCDs |
Large chlorhexidine gluconate carbon dots Medium chlorhexidine gluconate carbon dots Small chlorhexidine gluconate carbon dots |
DCFDA |
2’,7’-dichlorofluorescein diacetate |
DOX |
Doxorubicin |
DCDs |
Dual emission carbon dots |
EtOH |
Ethanol |
Fn-CDs |
F. nucleatum-carbon dots |
HBCUs |
Hypocrella Bambusa |
HBCDs |
Hypocrella Bambusa CDs |
GSH |
Glutathione |
GQDs |
Graphene quantum dots |
l-CDs |
Lotus bee pollen carbon dots |
MCDs |
Milk vetch-derived CDs |
NCDs |
Nitrogen-doped CDs |
NBCDs |
Natural biomass CDs |
NIR-CDs |
Near-infrared emissive CDs |
PDT |
Photodynamic therapy |
PL |
Photoluminescence |
PTT |
Photothermal therapy |
PCQDs |
Plum-based carbon quantum dots |
H2N-PEG-NH2 |
Polyethylene glycol diamine |
OCDs |
Osmanthus leaves-derived CDs |
QY, Ï• |
Quantum yield |
ROS |
Reactive oxygen species |
R-CDs |
Red-emitting CDs |
wCDs |
Rose-red fluorescence CDs |
SCDs |
Single-emission carbon dots |
MCDs |
Shiitake mushroom derived CDs |
TCDs |
Tea leaves-derived CDs |
Tf |
Transferrin |
- In the section on “Antibacterial activity”, information must be included on the cell toxicity of these CDs at the concentrations that are effective against bacteria (if available). Is there any insight on what are the functional groups on their surface that lead to these properties?
Answer: Thanks for your critical opinion. In the reference article, they mentioned that ‘‘CDs are internalized into bacteria. Also, the outer surface of bacteria is attached by CDs leading to indirect proliferating inhibition.”. The article also provides some supporting references (DOI: 10.2147/IJN.S121956, 10.7150/thno.39863, and 10.3390/nano11081877). Please kindly check the revised sentences in lines 351-353 and references with a red highlight.
References
- Int J Nanomed 2017, 12, 1227-1249.
- Theranostics 2020, 10, 671-686.
- Nanomaterials-Basel 2021, 11.
10. Several entries in Table 2 refer to CDs that are not derived from natural resources and therefore are not relevant.
Answer: We thank the reviewer for the detailed review. The irrelative references have been removed and transferred suitable works in this table. Please kindly check the revised table 2.
- The Discussion and conclusion section is very small and should be improved.
Answer: We thank the reviewer for the suggestion. We have added a paragraph on green chemistry in the section of “Discussion and conclusions.” Please kindly check the added paragraph in lines 431-442 with a red highlight.
Minor
- 8 in line 64, ref. 126 in line 180 and ref. 127 in line 185 are irrelevant; Also Ref. 80 in Fig. 9 and in the relevant text is not the correct one. I suggest to thoroughly check all references.
Answer: Thank you for your detailed review. The actual Ref. 80 is shown as follows: Li, Y.B.; Bai, G.X.; Zeng, S.J.; Hao, J.H. Theranostic Carbon Dots with Innovative NIR-II Emission for in Vivo Renal-Excreted Optical Imaging and Photothermal Therapy. Acs Appl Mater Inter 2019, 11, 4737-4744, doi:10.1021/acsami.8b14877. Please kindly check the revised references [157].
References
- Acs Appl Mater Inter 2019, 11, 4737-4744
2. Figure 1 mentioned in lines 224,228 is in fact Figure 2.
Answer: Thank you for your kind opinions. We corrected the mistake in lines 224 and 228; Figure 1 mentioned in this sentence means Figure 2 (a-f). Please kindly check the revised manuscript.
- Please explain the term “structural frustration” in line 290.
Answer: Thank you for your kind reminder. The typo has been revised to the structural properties in line 342-343.
- “SEM imagery” should be replaced by “SEM images”
Answer: Thank you for your kind reminder. The typo has been corrected in line 339.

Reviewer 3 Report
The review of M.-H. Chan and B.-G. Chen et al., titled "Natural carbon nanodots for theoretical evaluation and theranostic cancer application," describes carbon nanodots' in general, with a description of the preparation and application in cancer diagnosis and treatment.
The abstract and introduction need to be revised. After table 1, the manuscript is easy to follow with good explanations of each work. I recommend that the contribution may be published after major corrections that will be found in the PDF file attached and some questions that were raised:
- The title is confusing. Why theoretical evaluation? Several experimental data from other authors are compiled to give a complete vision of carbon nanodots in cancer.
Also, there is a section on antibacterial activity, so the title should reflect that you will generally talk about carbon nanodots. If not, please eliminate sections 3.2 and 3.3 that are not cancer-related.
- Some statements in the manuscript need references. They are marked in the PDF file.
- Line 90-91: 'Research has found that carbon nanoparticles synthesized by this method have good 90 quantum yield, although the quantum yield is not high.' What do you mean, good and not high? Is this related to stokes shift? Please extend the discussion.
- Line 98 says The production process does not require the use of any chemicals. The starting materials contain chemicals, and the process solvothermal, chemical ablation or pyrolysis, also requires chemicals.
- Figure 1 may be moved to line 159.
- You mention in the text several times that modifications of the CD's surfaces are easy. However, in the text, there are no examples.

Author Response
Response to reviewer 3:
Reviewer #3 (Reviewer Comments to the Author):
The abstract and introduction need to be revised. After table 1, the manuscript is easy to follow with good explanations of each work. I recommend that the contribution may be published after major corrections that will be found in the PDF file attached.
Answer: Thank you for your professional correction and detailed review. All the typos and mistakes have been revised in the manuscript. We are grateful for your kind guidance to improve our manuscript profoundly. Please kindly check the red highlight in the revised manuscript.
The title is confusing. Why theoretical evaluation? Several experimental data from other authors are compiled to give a complete vision of carbon nanodots in cancer. Also, there is a section on antibacterial activity, so the title should reflect that you will generally talk about carbon nanodots. If not, please eliminate sections 3.2 and 3.3 that are not cancer-related.
Answer: We are thankful for the reviewer’s suggestion. To make the title more consistent with all the content in the manuscript, we revised the new title: Natural carbon nanodots for its toxicity evaluation and theranostic biological application. Please kindly check the revised title for the manuscript.
Some statements in the manuscript need references. They are marked in the PDF file.
Answer: We are very grateful to accept the reviewers' corrections to this manuscript. According to these revisions, the narrative of the manuscript becomes more fluent.
Line 90-91: 'Research has found that carbon nanoparticles synthesized by this method have good 90 quantum yield, although the quantum yield is not high.' What do you mean, good and not high? Is this related to stokes shift? Please extend the discussion.
Answer: Thanks for your critical opinion. We are sorry that the description in this sentence is not precise enough. The corrected sentence is as follows: “Research has found that carbon nanoparticles synthesized by this method have good fluorescent performance, although the quantum yield is not high.” Please kindly check the revised Line 80-82 marked with a red highlight.
Line 98 says The production process does not require the use of any chemicals. The starting materials contain chemicals, and the process solvothermal, chemical ablation or pyrolysis, also requires chemicals.
Answer: We are thankful for the reviewer’s suggestion. The corrected sentence is as follows: “The production process does not require the use of many precursor chemicals to build the natural CDs.” Please kindly check the revised Line 89-90 marked with a red highlight.
Figure 1 may be moved to line 159.
Answer: Thank you for your kind suggestion. We have moved Figure 1 near to the description of the main application of CDs.
You mention in the text several times that modifications of the CD's surfaces are easy. However, in the text, there are no examples.
Answer: We appreciate the reviewer’s concerns. Based on the references in this manuscript, we list the modifications of the CD's surfaces in table 3. Please kindly check the new Table 3 mentioned for the surface modification.
Table 3. Modifications of CDs.
CDs type |
Modification |
Goal |
Ref. |
Carbon quantum dots |
Ethylene diamine |
Nucleoli selection |
[102] |
Nitrogen-doped carbon dots |
Folic acid |
Cancer cell targeting |
[105] |
Carbon dots |
Polyethylene glycol diamine; chlorin e6; transferrin |
Photosensitizing and cancer cell targeting |
[153] |
Carbon dots |
4-carboxy-benzyl boronic acid |
Tumor cell targeting |
[128] |

Round 2
Reviewer 2 Report
Dear Authors,
I believe that the manuscript has been sufficiently and that most of the comments have been addressed. I however believe that the new title is grammatically incorrect and requires your attention. Also the title “4.1. Bioimage” perhaps should be changed to “4.1. Bioimaging”. Finally, the Discussion section was very small and it was extended by adding a paragraph on green chemistry. The request was to expand the discussion and illustrate a future outlook on natural carbon dots, not on green chemistry in general which of course is useful but not to the point of this review.
Author Response
Dear Authors,
I believe that the manuscript has been sufficiently and that most of the comments have been addressed. I however believe that the new title is grammatically incorrect and requires your attention. Also the title “4.1. Bioimage” perhaps should be changed to “4.1. Bioimaging”. Finally, the Discussion section was very small and it was extended by adding a paragraph on green chemistry. The request was to expand the discussion and illustrate a future outlook on natural carbon dots, not on green chemistry in general which of course is useful but not to the point of this review.
Answer: We appreciate the reviewer’s comments. After checking with the title of the manuscript, we revise the title as: “Natural carbon nanodots: toxicity assessment and theranostic biological application.” In addition, the section title “4.1. Bioimage” has been changed to “4.1. Bioimaging”. Moreover, the “Discussion and conclusions” section has added the future outlook on natural CDs to discuss the challenges and future research goals. Please check the manuscript's revised title and section 4.1 and the revised “Discussion and conclusions” marked with red highlights.

Reviewer 3 Report
I suggest submitting this work after a minor revision:
Would you please revise the references in table 1? For example, reference 108 in the table should be 109, and so on.
Author Response
I suggest submitting this work after a minor revision:
Would you please revise the references in table 1? For example, reference 108 in the table should be 109, and so on.
Answer: We are thankful for the reviewer’s kind concerns. Sorry for the mistakes that we made in Table 1. We have rearranged the order of the references. Please kindly check the revised table 1 for the manuscript.
